# *Cronobacter* Species in the Built Food Production Environment: A Review on Persistence, Pathogenicity, Regulation and Detection Methods

**DOI:** 10.3390/microorganisms11061379

**Published:** 2023-05-24

**Authors:** Zeinab Ebrahimzadeh Mousavi, Kevin Hunt, Leonard Koolman, Francis Butler, Séamus Fanning

**Affiliations:** 1UCD-Centre for Food Safety, School of Public Health, Physiotherapy and Sports Science, University College Dublin, D04 V1W8 Dublin, Ireland; leonard.koolman@ucd.ie (L.K.); sfanning@ucd.ie (S.F.); 2School of Biosystems and Food Engineering, University College Dublin, Belfield, Dublin 4, D04 V1W8 Dublin, Ireland; kevin.hunt@ucd.ie (K.H.); f.butler@ucd.ie (F.B.); 3Department of Food Science and Engineering, Faculties of Agriculture and Natural Resources, University of Tehran, Karaj 6719418314, Iran

**Keywords:** *Cronobacter*, powdered infant formula, persistence, risk assessment, regulation, detection

## Abstract

The powdered formula market is large and growing, with sales and manufacturing increasing by 120% between 2012 and 2021. With this growing market, there must come an increasing emphasis on maintaining a high standard of hygiene to ensure a safe product. In particular, *Cronobacter* species pose a risk to public health through their potential to cause severe illness in susceptible infants who consume contaminated powdered infant formula (PIF). Assessment of this risk is dependent on determining prevalence in PIF-producing factories, which can be challenging to measure with the heterogeneity observed in the design of built process facilities. There is also a potential risk of bacterial growth occurring during rehydration, given the observed persistence of *Cronobacter* in desiccated conditions. In addition, novel detection methods are emerging to effectively track and monitor *Cronobacter* species across the food chain. This review will explore the different vehicles that lead to *Cronobacter* species’ environmental persistence in the food production environment, as well as their pathogenicity, detection methods and the regulatory framework surrounding PIF manufacturing that ensures a safe product for the global consumer.

## 1. Introduction

*Cronobacter* species are gram-negative rod-shaped bacteria, which are sized approximately 1 μm × 3 μm, and are members of the family *Enterobacteriaceae*. They express an oxidase-negative and catalase-positive phenotype, whilst being facultatively anaerobic, and are motile by peritrichous flagella [1]. *Cronobacter* species can grow in general bacteriological laboratory media, such as brain heart infusion (BHI) and tryptone soy agar (TSA), under temperatures ranging from 6 to 45 °C. Different temperatures have a significant effect on the growth of *Cronobacter* species in these media. When grown at room temperature, many of these bacteria can produce yellow-pigmented colonies, which is a phenotype that may challenge their correct identification. Current detection methods use a recommended growth temperature of between 37 and 44 °C. They reduce nitrate; consume citrate, hydrolyze esculin, and arginine; and decarboxylate L-ornithine. They are capable of metabolizing a number of mono-, di-, and tri-saccharides; sugar alcohols; and deoxy sugars, such as _D_-glucose, sucrose, raffinose, melibiose, cellobiose, _D_-mannitol, _D_-mannose, _L_-rhamnose, _L_-arabinose, _D_-xylose, trehalose, galacturonate and maltose.

*Cronobacter* species were formerly referred to as yellow pigmented *Enterobacter cloacae.* However, DNA hybridisation, biochemical reactions, pigment production and antibiotic susceptibility tests showed that what had previously been described as *Enterobacter cloacae* was in fact a different genus [2]. Further studies revealed that there were several species within this classification. Moreover, molecular approaches, such as fluorescent-amplified fragment length polymorphism (f-AFLP) analysis, automated ribotyping, full length 16S rRNA gene sequencing and DNA-DNA hybridisation, found a taxonomic relationship among 210 isolates of *Enterobacter* species [3]. Through evaluating the genetic relationship between these different isolates, a distinct genus, named *Cronobacter* species, was proposed [4]. Phenotypic analysis of *Cronobacter* species revealed differences between them [5,6,7]. These phenotypic data, subsequently supported via both optical mapping and genome sequencing, finally resulted in the revision of the bacterial taxonomy [8,9,10]. Originally, six species within the new genus *Cronobacter* (*C. sakazakii, C. malonaticus, C. turicensis, C. muytjensii, C. dublinensis* and *C. universalis*) were defined, comprising the 16 biogroups described in Table 1. The division of species is, for the most part, equated to particular previously described biogroups [10,11,12]. A seventh species, named *C. condimentii*, was later included.

Among *Cronobacter* species, both *C. sakazakii* and *C. malonaticus* are closely related, as determined via 16S rRNA sequencing, and these species could not be discriminated using 16S rRNA gene sequence analysis [13,14,15,16]. Subsequently, a multi-locus sequencing typing (MLST) scheme was developed to distinguish between these and all other isolates within the genus [17,18]. Further studies reported on the epidemiological importance of sequence type (ST4) within *C. sakazakii*, which is responsible for a large proportion of the documented severe neonatal infections, especially neonatal meningitis [19].

This review considers the advances in our understanding of pathogenicity mechanisms of *Cronobacter* species in the food production environment, public health and novel approaches to detection and tracking across the food chain.

## 2. Public Health Significance of *Cronobacter* Species as a Bacterial Hazard

*Cronobacter* species is a recognised risk to public health arising from its potential to cause severe illness in susceptible infants exposed to contaminated PIF [20,21]. From a risk management perspective, infant feeding is of most concern during the early months of life when an infant’s immune system is underdeveloped and PIF is used as an alternative to breastfeeding [22]. *Cronobacter* is not generally associated with breast milk consumption [23,24], except in cases of improperly sanitised breastfeeding equipment [25,26]. The risk of illness associated with *Cronobacter* in vulnerable demographics makes control measures necessary during production of PIF [27,28].

### 2.1. Impact on Public Health

PIF contamination is responsible for most reported cases of *Cronobacter* in infants. Usually, these events start with a batch of defective PIF that was contaminated during production through either the use of contaminated ingredients or cross-contamination. Food-borne *Cronobacter* case rates are likely to be under-reported, which is similar to other infectious diseases. The rate of under-reporting are low, however, given the general severity of outcomes in infant populations. Mead and Slutsker [29] estimated an under-reporting ratio of 2 for the United States, meaning reported hospitalisations represent 50% of actual hospitalisation cases. Reij and Jongenburger [30] estimated that *Cronobacter* infection represents 0.5 to 2.4% of the total foodborne disease burden and 0.5–0.7% of the meningitis burden. Many recall events for PIF were previously carried out in different global production regions after *Cronobacter* contamination was discovered [31,32], including a major event in February 2022 in the United States [33]. In the EU and the US, recalls and safety alerts are reported by RASFF and the FDA, respectively.

Table 2 highlights all accessible recalls and alerts for the reporting period 2018–2021; these data were obtained from sources in the EU [34] and the US [33]. Both *Cronobacter* and Enterobacter species data are included. Combined, there were 20 alerts in the EU across all food types and 6 in alerts the US. In comparison, there were 1130 total alerts for *Salmonella* in the EU for the same reporting period and 183 alerts in the US. In total, 72.2% of the EU *Enterobacter* alerts were associated with animal feed and pet food.

### 2.2. Defining the Population at Risk

*Cronobacter* infection is possible in all demographics; however, it is most likely in populations that combine lower immune status and consumption of rehydrated milk, i.e., new-born infants and certain elderly subpopulations [34]. Infants under the age of 12 months were previously identified as most at risk for *Cronobacter* infection following consumption of contaminated PIF [20]. Based on a risk assessment, subpopulations of further interest are new-born infants under the age of one month and all infants of low birth weight under the age of two months [12,20,30]. The post-infection mortality rate in these groups ranges from 20 to 80% [35]. Until 2002, the majority of *Cronobacter* outbreak events occurred in these subpopulations; in that year, health agencies began recommending sterile liquid formula for hospitalised infants under the age of two months [36].

#### 2.2.1. Neonatal Infections

Although all *Cronobacter* species were isolated from clinical specimens, investigations showed that most infections were caused by *C. sakazakii*, *C. malonaticus,* and *C. turicensis* [15]. Clinical symptoms of *Cronobacter* infections generally include necrotizing enterocolitis (NEC), septicaemia and meningitis in neonates and infants, compared to wound and urinary tract infections, septicaemia, vaginitis and aspiration pneumonia in adults [34,35]. Most of the outbreaks due to invasive *Cronobacter* infections among infants reported over the years were attributed to PIF that was either intrinsically or extrinsically contaminated with *C. sakazakii* [37]. Furthermore, there are several reports of infants being infected through contaminated expressed breast milk and cross-contamination from improperly sanitised breast milk pumps [34,38,39]. Premature infants, low-birth-weight neonates and infants with underlying medical conditions are at the highest risk for developing severe *Cronobacter* infections. Mortality rates among this group were previously reported to be approximately 27%, and infants who survive often suffer developmental delays, hydrocephaly, learning disability and other neurological sequelae. In addition, [35] concluded that, apart from health care standards regarding both the therapeutic methods and the infection control procedures, regional (continental), seasonal and climatic or genetic variations need the most consideration regarding *Cronobacter* species infections. Using multi-locus sequencing typing (MLST), it was suggested that *C. sakazakii* strain ST4 is predominantly associated with illnesses in infants and children [9,40].

#### 2.2.2. Infections in Susceptible Older Adults

Studies revealed that healthy adults are considerably less susceptible to infection with *Cronobacter* species [41]. However, immunocompromised adults and the elderly with aged above 80 years old could be susceptible to the bacterium [39,42]. Additionally, the risk of *Cronobacter* infection in elderly adults who have experienced stroke is high. This group of people have diminished swallowing abilities (dysphagia) [43]. Therefore, consumption of rehydrated powdered protein supplements as part of their diet and rehabilitation efforts could be a source of *Cronobacter*.

Unfortunately, because the reporting of *Cronobacter* infections is not mandatory in many countries, the true incidence of invasive infant *Cronobacter* infections, as well as adult infections, are unknown [44].

### 2.3. Quantitative Risk Assessment (QRA) for Cronobacter in PIF

Due to the absence of reliable dose-response models, initial risk estimates for *Cronobacter* in PIF could not be assessed in absolute terms [30]. Risk assessments were carried out by the FAO/WHO [20,45], mainly through comparing scenario analyses. These were limited due to the absence of dose-response data and the deterministic modelling of uniform cell consumption assumption. In France, one risk assessment estimated 0 and 3 cases per 100,000 infants in the first 6 months of life [22].

#### 2.3.1. Infectious Dose

The median infectious dose for food-borne *Cronobacter* is unknown and is likely to vary depending on the nature of the isolate [46]. The lethal dose for intravenous *Cronobacter* in mice was estimated at 108 CFU/mouse [47]. An approximate foodborne median infectious dose of 1000 CFU was proposed by Iversen et al. and supported based on oral challenge studies in mice and rats [48,49]. A risk assessment by Boué and Cummins [22] used an exponential dose-response model, with the single parameter ‘r’ varying from 10^−5^ to 10^−10^ and not dependent on infant age. [46] suggested that potential exposure to *Cronobacter* the day before onset of illness was between 2160 and 3600 CFU.

#### 2.3.2. Transmission Pathways and Exposure Routes

Contaminated food is the main transmission pathway for infant exposure to *Cronobacter*. After ingestion, the infection can pass outside the gastrointestinal system and lead to meningitis [21]. PIF is contaminated with *Cronobacter* extrinsically during factory production and intrinsically through the ingredients used [50]. PIF is not a sterile product and other possible pathways of transmission arise from cross-contamination during rehydration, which often occurs due to inadequate cleaning or handling practices [22,45]. Surfaces, utensils and factory equipment can act as reservoirs for *Cronobacter* during PIF production [51] Control at the factory stage is most effective means of reducing risk [27].

#### 2.3.3. Sampling Protocols

When sampling for *Cronobacter*, the estimated bacterial numbers will vary depending on how much clumping or clustering is present or assumed in the matrix. Incorrect assumptions of homogeneity will lead to inaccurate estimates of concentration [52]. This issue is especially relevant in the absence of dose-response data since total prevalence and consumption are the main drivers of risk analysis [30]. The pattern of contamination will depend on its source and frequency. FAO risk assessments assumed a homogenous Poisson distribution of *Cronobacter* contamination in PIF [45,53]. This assumption was shown to be insufficient, as demonstrated by other authors [54,55,56]. *Cronobacter* CFU are more likely to occur in clusters, where the contents of each cluster can be represented via a lognormal or gamma distribution and the clusters themselves are Poisson distributed. They, therefore, require compound distributions, such as the Poisson–lognormal or Poisson–gamma, to be modelled accurately [57,58]. This clustering of *Cronobacter* in PIF can have a significant effect on exposure assessment modelling. If exposure assessment assumptions are inaccurate, they can lead to misinterpretation of sampling results (positive or negative) and, therefore, over- or under-estimation of risk [56,59]. Exposure events may be higher or lower than predicted, and bacterial cells may be more likely to survive desiccated conditions. For *Cronobacter*, prevalence rates of between 3 and 30% are reported throughout the literature, varying depending on the region [15,60,61,62]. Positive prevalence shows the need for continuing control measures, given the risk posed to the infant population. In the United States, an infection rate of 1 in 100,000 new-borns was previously estimated, representing 0.53 infections per year [63].

#### 2.3.4. Total Consumption and Global Exposure

Reij and Jongenburger [30] estimated global exposure to *Cronobacter* through multiplying the estimated global PIF consumption [64] by the estimated *Cronobacter* prevalence [45]. The result was 7.84 log_10_ CFU/year in total, not including post-rehydration growth. The accuracy of this estimate is contingent on having accurate estimates for concentration and spatial distribution. More complicated and specific estimates for consumption are demonstrated in other published risk assessments [22], which estimated a global exposure rate of 0.33 MPN CFU/g.

### 2.4. Control Measures in the Context of Public Health

As mentioned above, PIF is not a sterile product [45,65]. Control and safety of PIF is best maintained early in the production process. This method can be undertaken through implementing standard practices for hygiene—GHP (Good Hygienic Practices), GMP (Good manufacturing practice) and HACCP methods [66]. Methods to reduce water in the factory environment are described in FAO/WHO guidance documents. Over time, incidences of *Cronobacter* infection decreased [44]. The points of transmission come both during production and consumer rehydration. Control measures for *Cronobacter*, therefore, focus on both manufacturing and reconstitution. HACCP programs are used to manage contamination, though are not mandatory for infant formula producers under Codex guidelines. GMP practices are regulated, and general hygiene and handling protocols are effective in minimising *Cronobacter* risk [27,67]. Codex have set out microbiological criteria for *Cronobacter* contamination that can be used by national regulatory agencies. There are separate criteria for products intended for infants under the age of 6 months and those for young children aged 12 to 36 months [27,45]. Hygienic control measures and microbiological criteria are described in FAO and WHO documents [27]. For consumers, the control steps for *Cronobacter* risk mitigation include the use of sterile liquid formula where appropriate, rehydration at temperatures above 70 °C and minimising time between preparation and consumption [20,21]. *Cronobacter* can grow during rehydration and reach infectious doses starting from lower concentrations. Rehydration at temperatures close to 35 °C supports rapid growth, with 1 log_10_ increases observed in 1 to 5 h [46].

In summary, *Cronobacter* poses a threat to infants under the age of 12 months through the consumption of non-sterile powdered infant formula. Assessment of this risk is dependent on accurate estimation of prevalence in factories, which can be challenging to measure due to the heterogeneity observed in factories [41]. There is also potential risk of growth during rehydration, given the observed viability of *Cronobacter* in desiccated conditions.

## 3. Global Manufacture and Trade of PIF

The powdered formula market is large and growing. It includes at least three distinct categories of product, each with a specific definition and market share. *Codex Alimentarius* defines PIF as a powdered milk product intended as a breastmilk substitute for infants aged up to 6 months old [20]. Follow-up formula (FUF) is defined by Codex as “a liquid part of the weaning diet for the infant from the 6th month onwards and for young children” [27]. There are also guidelines for special formula, often intended for medical purposes, as a separate category of product [20]. Euromonitor International, which is a private company that is commonly cited in market estimations, tracks formula products under four categories: standard (0–6 months), follow-up (6–12 months), toddler (13–36 months) and special (0–6 months) [68]. Some aggregated data on dairy manufacture and trade are available from agencies such as Eurostat, the WTO and the FAO. The most reliably cited source for PIF retail sales and manufacture was Euromonitor International. These data are only available under license, though Baker and Santos [68] provide summaries of Euromonitor formula data for the period 2005–2019. These data represent the most recent and comprehensive public figures available, as set out below.

### 3.1. Sales and Consumption of PIF

#### 3.1.1. Global Market

Various estimates of the size of the global formula market are available from different sources in the literature. In 2013, it was estimated to be valued at USD 41 billion [69]. By 2015, this value had increased to USD 47 billion and WAS forecast to grow to between USD 60 and USD 70 billion by 2021 [70]. In 2019, the market was projected to reach over USD 100 billion by 2026, with the United States accounting for only 5% of demand [71]. According to Euromonitor International, total milk formula production was 0.97 million tons in 2005; this figure increased to 2.15 million tons in 2019, with growth of 10% to 2.38 million tons predicted by 2024 [68]. As of 2019, global sales of standard PIF (ages 0 to 6 months) were 24% of the total formula market, sales of follow-up formula (6–12 months) were 22%, sales of growing up or toddler formula (12–36 months) were 48% (reflecting the higher population in this group), and sales of special formula (0–6 months with additional medical requirements) were 5.6% [68]. In 2018, the proportion of infants under the age of six months exposed to some proportion of PIF in their diet was approximately 60% [72].

#### 3.1.2. Trends in Sales and Manufacture

Sales and manufacture of PIF more than doubled between 2005 and 2019, with a 120% increase from 3.5 to 7.4 kg/infant globally [68]. This growth was largely driven by emerging economies, in particular China. Compared to high-income countries, emerging markets make up most of the global growth in infant formula, with China contributing more than half of this growth alone [69]. China is a significant global market for infant formula because of its size, its growing middle class and the history of unsafe domestic PIF use [73]. In 2008, Chinese-manufactured PIF was found to be contaminated with melamine, which had been deliberately added to disguise poor quality ingredients. This scandal led to increased demand for imported formula. China represented 14.1% of the global market in 2005, increasing to 32.5% by 2019 [68]. The rapid growth rates seen for demand in China in recent years are forecast to reduce, though they will still increase by more than the comparative figures for the EU and the US. This trend is partly due to falling birth rates. China remains the biggest single market for PIF, with a low rate of breast feeding, which is forecast to fall further as economic development continues [74].

As of 2015, over 60% of global PIF was produced by the top six manufacturers: Nestlé, Danone, Mead Johnson, Abbott, Friesland Campina and Heinz [70]. Many smaller competitors also operate in joint ventures with these companies [69]. In 2017, the US FDA estimated that there were 40–50 active formula processing plants worldwide [75]. According to Kent [69], production scaled up in response to increasing demand through the use of joint business ventures and centralisation of the production system. The three biggest production regions for PIF, accounting for over half of production, are the EU, with approximately one third of the global total; China; and South-East Asia (SEA). Of the other production regions, the US, New Zealand and South America are most significant [74]. The highest PIF-producing countries in the EU are France, the Netherlands, Ireland, Germany, Spain and Poland. The highest producing countries in SEA are Indonesia, the Philippines, Thailand, Vietnam and Singapore. A 2017 report on the industry published by Gira Foods [74] showed the EU to be a net exporter, exporting approximately 60% of its total production. EU exports grew at a rate of over 10% per year from 2012 to 2021. About half of EU exports went to China, the biggest single market, with 24% going to the Middle East–North Africa region, and 5% going to Southeast Asia. As a region, South-East Asia was a net importer, with 15% of its consumption imported from the EU and other regions. China was also a net importer of PIF and exported little of its domestic production. Partly due to the earlier melamine contamination scare, in China, there is high demand for imported infant formula.

## 4. Regulatory Guidance and Legislation for PIF

The regulatory framework that exists for PIF around the world addresses the immediate safety and composition challenges presented by the product. In 2004, the FAO/WHO report concluded that present standards and technology could not remove the risk of contamination through the application of regulation or manufacturing guidelines [20]. This issue means that PIF is not sterile, even with the highest manufacturing standards, and additional guidance is needed at the point of preparation [21]. International guidelines state that labelling should contain sufficient direction for safe consumption of the product [45]. Different countries can, however, have more stringent labelling requirements [20,76].

In general, food safety regulation is managed at a national or subnational level. Food is produced and traded globally, which creates the need for guidance at the global level, in particular for those importing countries that depend on external production [77]. Countries can choose to implement the Codex Alimentarius Commission (CAC) guidelines through their own national or regional agencies. In developing countries, PIF is imported from processing plants based in countries such as Bangladesh, where contamination status is unlikely to change in transit, making international standards for production the most significant safety guarantor [20]. Not all production countries follow the same CAC definitions for PIF: for example, recommended age boundaries vary between the US and the EU [27].

### 4.1. Codex Framework

Since the 1970s, the CAC has produced several guidelines and amendments on the safety and ingredients of PIF (CXA 2-1976). The Codex Code of Hygienic Practices for Foods for Infants and Children was the original source for guidance and regulation, followed by later amendments [20]. In 2008, CAC adopted the Code of Hygienic Practice for Powdered Formulae for Infants and Young Children (CXC 66-2008), which included guidance on microbiological criteria [27]. In these guidelines, the term “powdered infant formula” refers to products manufactured in powdered form, which are intended for consumption by infants. This includes infant formula, follow-up formula and special medical formula [20].

#### 4.1.1. Developments and Trends in Guidelines

Codex was advised by the FAO/WHO in 2004 to update its guidelines for PIF safety, which, in practice, means establishing microbiological criteria for *Cronobacter*. Part of these recommendations were directed towards manufacturers, both to facilitate their implementation of environmental monitoring, and to use *Cronobacter* species presence as indicator bacteria during production, instead of coliforms [20]. FAO/WHO risk assessments [45] provided guidance to Codex and other stakeholders on introducing microbiological criteria to reduce risk, particularly regarding the sampling plan. This approach led to updated Codex guidelines in 2008, which were drafted by a working group headed by Health Canada [21]. *Cronobacter* species was proposed as a new genus in 2008, reclassifying those earlier isolates previously recognised within the genus *Enterobacter*. This revision did not affect the defined organisms or the regulatory framework [27,78].

#### 4.1.2. Standards for Manufacture of PIF

PIF is produced through the “wet blend” method, the “dry blend” method or a combination of the two methods. In wet blending, a wet mixture of ingredients is combined and then dried via heating. The heating step aids elimination of bacteria. Dry blended powders do not include this process step, instead blending dry raw ingredients. Combination processes apply wet blending to some ingredients but not to others [79]. Strategies for reducing *Cronobacter* in PIF during manufacturing discussed in the FAO/WHO meeting report of 2004 included guidelines on managing environmental hygiene, monitoring finished products and defining tighter microbiological specifications for PIF [79]. During production, a combination of HACCP and GMP are recommended to control *Cronobacter*, particularly in relation to equipment and water management. The use of Performance Objectives and Environmental Monitoring are also key recommendations [66]. Codex guidelines (CAC/RCPI-1969 REV.3 1997) advise the implementation and enforcement of GMP and the General Principles of Food Hygiene [80]. The use of these practices is described in guidance from Codex and other international organisations, along with guidance on environmental monitoring [20,27,45].

#### 4.1.3. Microbiological Criteria

In 2007, the Code of Hygienic Practices was revised to add criteria for *Cronobacter*, allowing a maximum of zero positives in 30 samples of 10 g of product [27,81]. The risk assessments of the FAO/WHO groups indicate that stricter microbiological criteria would not be more effective in reducing risk [20,64].

### 4.2. National Specifications in Formula for Infants

#### 4.2.1. European Union

The EU defines infant formula as food intended for children under the age of 12 months, in the absence of complementary feeding, and follow-on formula as reconstituted liquid intended for children aged 12 to 36 months as part of a diversified diet. Other definitions, such as growing up milk, are not defined in EU law [82]. PIF standards are regulated through Directive 2006/141/EC for standard and follow-on formula and Directive 1999/21/EC for special medical formula. The microbiological criteria for *Cronobacter* in PIF production are established in the EU in Regulation (EC) No 2073/2005 and are the same as those recommended in Codex guidelines. The detection method specified in the regulation is presently ISO 22964:2017 [83], which uses non-selective culture enrichment.

#### 4.2.2. United States

In the United States, PIF is regulated through the Infant Formula Act (21 U.S.C. 350a) and related statutes (21 CFR 106 and 107). These regulations define safe manufacturing based on GMP, including microbiological controls, as well as nutritional and labelling requirements. These acts include regulations on sanitation and construction of production facilities and processes [80]. Specific controls for infant formula are defined within the FDA’s definitions of GMP [84]. US standards do not overlap entirely with those of the EU, and some EU imports were previously found not to be compliant with FDA labelling requirements [85]. In 2014, the FDA updated the manufacturing standards for safe production to include routine testing for *Cronobacter* and *Salmonella*, which corresponded with lower incidence rates [44].

#### 4.2.3. Canada

In Canada, manufacture of PIF is also subject to GMP guidelines, which are published by the Canadian government. The import standards for PIF in Canada are that the product must be manufactured in accordance with the Canadian GMP guidelines [21]. HACCP is not mandatory, though encouraged via GMP application [80]. Codex microbiological criteria for *Cronobacter* are also in place in Canada [86].

#### 4.2.4. China

A series of food safety incidents in China affecting PIF that occurred between 2005 and 2012 lowered confidence in Chinese-produced PIF, giving rise to a major legislative response [87]. The most notable of these incidents was the largescale and deliberate contamination of PIF with melamine to conceal nutritional deficiencies [69]. China introduced several new laws enforcing food safety standards following these events. The main directive controlling PIF production is the Food Safety Law, which is enforced by the recently established Food Safety Commission of China [88]. GMP has been mandated in Chinese manufacturing plants since 2010 (GB 23790-2010). Several additional standards and practices for PIF manufacture were introduced, and investments in dairy quality production and inspection were increased [87].

## 5. Phenotypes Expressed by *Cronobacter* Species Contributing to Their Persistence in a Production Environment Setting

In the modern production setting, various measures are deployed to manage the risk of *Cronobacter* species colonisation. Implementation of strict hygiene measures, along with other controls, are important as a food safety measure. Nonetheless, the bacterium can respond to these challenges in various ways; some of these counter measures are outlined below.

### 5.1. Biofilm Formation

The biofilm formation of *C. sakazakii* in different ecological settings, including food production environments, hospitals, day-care centres and other environments, was previously reported [89,90,91,92]. These organic matrices, which are composed of heteropolysaccharides [93,94], can lead to an increased resistance to environmental stresses, as well as to antibiotics and detergents [95]. It was previously reported that benzalkonium chloride (BAC), which is the most commonly used member of the quaternary ammonium compounds (QAC) family of disinfectants, could result in an increase in the disinfectant tolerance in this bacterium [96,97,98]. Significant increase in *Cronobacter sakazakii* survival in the presence of BAC could trigger biofilm formation as a counter measure designed to protect the bacterium from environmental stresses [99,100]. This phenotype increases the risk for onward transmission, including infection, in susceptible individuals [101,102]. Biofilm formation was previously reported on a range of environmentally relevant surface matrices, such as silicon, glass, stainless steel, latex and polycarbonate surfaces. Stainless steel was previously recorded as producing a smaller viable bacterial count after biofilm formation when compared to silicon, latex and polycarbonate [103,104].

Various genes play a functional role in biofilm formation in *C. sakazaki*. These genes include those encoding the biosynthesis of colonic acid, flagellar assembly protein (FliH) and flagellar protein (FlgA-K). *FlhA-E* and *flgJ* are popular genes linked to the biosynthesis of flagella [105]. Flagella contribute to cell motility, which is very important during biofilm formation and bacterial dispersal from a fully formed biofilm, as well as sensing and colonisation on different surfaces [106].

### 5.2. Thermal Tolerance

Different environmental factors, such as temperature, humidity, process conditions and cleaning schedules, could all influence the growth and survival of bacteria and potentially induce an adaptive response. Thermal processing methods used in dairy plants, such as spray drying, are not intended as a kill step to eliminate bacteria but could induce an adaptive response, increasing survival at elevated temperatures. Several studies reported on the heat tolerance of specific strains of *Cronobacter sakazakii* [107,108,109,110,111,112]. Environmental factors, such as pH, water activity (a_w_) and heat shock, could affect the thermal tolerance of *Cronobacter* species. In early studies on thermal inactivation of *Cronobacter sakazakii*, *D*-values in the range 3.44–5.45 min at 58 °C were reported [112]. Breeuwer and Lardeau [107] reported a lower D-value range of 0.3–0.6 min at 58 °C. Dalkilic-Kaya and Heperkan [113] investigated *C. sakazakii*, *C. dublinensis* and *C. muytjensii* for their thermal inactivation. *C. muytjensii* had the highest heat tolerance when measured at lower temperatures. The *D*-values at 52 and 54 °C were 33.30 (±1.17) and 6.79 (±1.05) minutes, respectively. At higher temperatures *C. sakazakii* exhibited the highest *D*-values for some of the study isolates. *D*-values at 56 and 58 °C were 4.73 (±0.40) minutes for one isolate and 2.30 (±0.26) minutes for another, while a further two other *C. sakazakii* had *D*-values of 1.17 (±0.03) and 1.14 (±0.02) minutes. Regarding the effect of heat shock on *Cronobacter* species, various results were reported. Arroyo and Cebrian [114] recorded a maximum heat resistance in *Cronobacter* species after they were incubated at 20 °C.

Other *D*-values were reported by different research groups that were supposed to be related to different types of cultivation media, different bacterial growth phases or different heat shock treatments used. These different D values demonstrated that there was a potentially large strain-to-strain variation in the thermal tolerance of *Cronobacter sakazakii* [115,116]. Through increasing pH from four to seven, a ten-fold increase was measured in thermal resistance for *Cronobacter* species [117,118,119,120]. A decrease in a_w_ from 0.99 to 0.96 could similarly result in a thirty-two-fold increase in the thermal resistance of the *Cronobacter* species at 4 °C and pH 4 [102].

Previously, an 18 kbp DNA region in some *Cronobacter* genomes was identified as contributing to prolonged survival at 58 °C. This 18 kbp region, which contained 22 open reading frames sizes ranging from 141–2850 bp, was sequenced in *Cronobacter sakazakii* ATCC™29544. The major feature of the region contained a cluster of conserved ORFs (denoted as *orf*A-Q), most of which had significant homologies with bacterial proteins involved in some type of stress response, including heat, oxidation and acid stress [121]. In addition, Nguyen and Harhay [122] examined heat tolerance islands across a broad range of bacterial species, including *Salmonella* and *Cronobacter.* They established three lineages with varying numbers of ORFs. *Cronobacter sakazakii* strain ATCC™29544 with 22 ORFs was found to be part of lineage two. In addition, *C. sakazakii* SP291 was characterised by a shorter (6.1-kbp) DNA region and identified to be part of lineage three. This work demonstrated that two versions of a thermotolerant DNA region can be found in *Cronobacter sakazakii*. Both islands contained a heat shock protein (hypothetical protein), followed immediately by a Clp protease. These proteins may play an important role in conferring thermal tolerance to the bacterium.

A proteomic approach for identifying markers associated with thermal resistance in strains of *Cronobacter sakazakii* was reported by Williams and Monday [123]. In this study, the Mflag020121 protein was found in all thermal tolerant strains and was identical to a protein found in the thermal tolerant bacteria: *Methylobacillus flagellatus* KT.

### 5.3. Acid Tolerance

*Cronobacter* species can be described as moderately acid-resistant bacteria [102,124]. The survival rates previously measured for *Cronobacter* species cultured from acidic food products vary [120,125]. At an incubation temperature of 25 °C, *Cronobacter* species were reported to grow in tomato, watermelon, and cantaloupe juice (all pH > 4.4) [126,127]. These bacteria did not grow in strawberry and apple juice wherein the pH was lower (both pH < 3.9) [126]. Survival of *Cronobacter* species was also recorded in fermented foods and food products with a neutral pH value [102,126,128].

### 5.4. Osmosis and Desiccation

*Cronobacter* species were shown to survive in foods with a low water activity (a_w_ 0.25 to 0.69), such as powdered infant formula (PIF) and infant rice cereals [129,130,131]. The bacterium is known to survive for up to two years in a desiccated environment and can multiply rapidly once rehydrated [102,107]. The ability of the *Cronobacter* species to survive in these conditions is linked to the intracellular accumulation of compatible solutes, such as trehalose, during the stationary phase [107,111]. This process leads to the development of desiccation tolerance arising from the stabilisation of proteins and phospholipid membranes. In general, *Cronobacter* species were shown to be more tolerant to osmosis and desiccation stresses than *E. coli* and some *Salmonella* species [102,107]. Several genes were reported to be involved in tolerance to various stressful environments, such as desiccation [132]. Genes encoding the osmotically inducible protein (OsmY), transcriptional regulatory protein (YciT), aquaporin Z, hyperosmotic potassium uptake protein (TrkH) potassium uptake protein (TrkA and TrkG), ProP, betaine aldehyde dehydrogenase (OtsA), Trehalose-6-phosphate hydrolase (OstB) and glutathione-regulated potassium-efflux system protein (KefB, KefC, and KefG) contribute to *Cronobacter* species survival in high osmotic environments [132].

It was reported that in heat/cold shock stress, DnaJ and DnaK suppressor proteins and the heat shock proteins (YciM) and (GrpE) are expressed [133,134]. In this situation, the cell rapidly accumulates electrolytes to increase the internal osmotic pressure in desiccated environments [132,134].

## 6. *Cronobacter* Species Pathogenicity

*Cronobacter* demonstrate a variable virulence phenotype depending on the nature of the isolate [135,136,137]. Currently, the precise mechanism via which *Cronobacter* expresses its virulence phenotype remains largely unknown [137]. Data from experimental animal models of meningitis and of tissue culture assays were reported in [138]. Studies using a new-born rat infection model suggested that enterocyte apoptosis, which is controlled through the induction of high levels of nitric oxide (NO) synthase, may be responsible for necrotizing enterocolitis (NEC) [49,139].

In addition, genome sequencing highlighted potential markers responsible for virulence in *Cronobacter* species [39,140]. Virulence factors contributing to *Cronobacter* pathogenicity consists of genes that function in both adhesion and invasion [40]. A diverse range of known virulence factors can be grouped into flagellar proteins, outer membrane proteins (*ompA*), chemotaxis (*motB*), hemolysins (*hlyIII*), invasion (*lpxA*), plasminogen activator (*cpa*), colonisation (*mviM*) and a transcriptional regulator (*sdiA*), macrophage survival, sialic acid utilisation (*nanA*, *nanK*) and toxin biosynthetic processing factors [40]. In addition, resistance to beta-lactam antibiotics, such as cephalothin, cefotaxime, ceftazidime and ampicillin, could assist in *Cronobacter* pathogenicity [141]. It was reported that the overuse of antibiotics in food environments and the presence of several antibiotic resistance operons (*marA*) can favour the development of resistance to different antibiotics in *Cronobacter spp.*

The mechanism of action for each virulent factor is different. For instance, genes encoding flagellar proteins function for bacterial motility, adherence capacity, biofilm formation and stimulation of pro-inflammatory responses through Toll-like receptor 5(TLR5) signalling [142]. *zpx* gene-encoding proteolytic enzymes lead to the deformation of host cells. These enzymes can invade capillary endothelial cells; persist in human macrophage, wherein it can influence cytokine secretion; and induce sever brain pathology in the neonatal rat [136,143,144,145]. The OmpA and OmpX proteins of *C. sakazakii* could contribute to basolateral adhesion and invasion to Caco2 and INT-407 cell lines, in addition to a possible involvement in the crossing of the blood–brain barrier (BBB). Genes such as *nanA* and *nanK* function to contribute in sialic acid consumption as a carbon source [40,134]. Studies reported an evolutionary adaptation of *C. sakazakii* to this compound, which is naturally found in breast milk and is artificially added as a supplement to powdered infant formula (PIF) due to its association with brain development, as it is a major component of gangliosides [40,146]. Regulation of the expression of enzymes such as sialidase and adhesins or inhibiting transcription factors of the *fimB* gene, which is part of the *fim* operon, u sialic acid could modify bacterial surface properties, mediating cell adhesion and invasion [40,147]. Many pathogenic bacteria use sialic acid to decorate their cell surface, which results in important phenotypic traits regarding their ability to interact with host cell surfaces and tolerate host innate immune responses [147].

## 7. Sources of *Cronobacter* Species Contamination

*Cronobacter* demonstrate a variable virulence phenotype depending on the nature of the isolate [135,136,137]. Currently, the precise mechanism through which *Cronobacter* expresses its virulence phenotype remains largely unknown [137]. Data from experimental animal models of meningitis and tissue culture assays were reported in [138]. Studies using a new-born rat infection model suggested that enterocyte apoptosis, which is controlled through the induction of high levels of nitric oxide (NO) synthase, may be responsible for necrotizing enterocolitis (NEC) [49,139].

### Powdered Infant Formula (PIF)

*Cronobacter* species can be isolated from foodstuffs other than those foods listed above. Although no known cases of illnesses arising from *Cronobacter* species were reported as arising from non-PIF foods, this food-borne pathogen is primarily associated with PIF and is considered a public health risk in this context [42,60,148,149,150,151]. All steps involved in PIF processing, including the raw material preparation, concentrate storage, mixing, drying, agglomeration and filling steps, affect the prevalence of *Cronobacter* in PIF production. Findings show that locations close to the filling of the final product and the intermediate stage of process have higher risk of product contamination. Investigations highly recommend correctly installing the air filters to reduce the dissemination of *Cronobacter* and other biological hazards in the food production setting. Many food recalls arise due to contamination of PIF with *Cronobacter* species; these recalls occurred in various countries over the years. Recent recalls were reported from Canada and the United States [152,153]. Contamination of PIF with *Cronobacter* species and the management of risk to consumers is a challenge to public health, regulatory agencies and manufacturers alike [60,154].

As mentioned before, *Cronobacter* species, including *C. sakazakii*, can survive in dry environments and the bacterium expresses a higher tolerance to osmotic stress and desiccation compared to other members of the *Enterobacteriaceae* family [155,156]. *C. sakazakii* was isolated from raw material and PIF processing environments (including roller dryers, drying towers and tanker bays, as well as floors and soil adjacent to production facilities) and shown to persist in these environments for long periods of time due to its resistance to desiccation and ability to survive spray drying [3,42,157,158,159].

## 8. Current Detection Methods for *Cronobacter* Species

Several detection strategies were developed and validated and specifically designed to detect *Cronobacter* species. These detection strategies are based on culture techniques. PCR, immunological and/or biosensor-based methods. A summary of these methods, including their specificity and time to detection, is provided in Table 3.

### 8.1. Culture-Based Detection of Cronobacter Species

Violet Red Bile Glucose agar (VRBGA), in combination with tryptic soy agar (TSA), is recommended by the US Food and Drug Administration (FDA) for the isolation of *Cronobacter* species [182]. In this method, enriched samples are plated onto VRBGA and presumptive colonies are sub-cultured onto TSA and incubated for 48–72 h at 25 °C. Yellow pigmented colonies on TSA are then confirmed biochemically using API 20E biochemical galleries [163,164]. This method requires a long testing period (5–7 days); thus, additional culture-based methods, such as selective, differential or fluorogenic-based agars, were developed for the detection of *Cronobacter* species. Fermentation of sucrose and metabolism of 5-bromo-4-chloro-3-indolyl-ɑ-ᴅ-glucopyranoside (X-gal) can be used to distinguish *Cronobacter* species and select for growth. Chromogenic and fluorogenic agars are based mainly on the enzyme ɑ-aminoglycosidase and its fluorescent-based substrates, which can be used as markers for the presence of *Cronobacter*. For example, 4-methylumbelliferyl-ɑ-ᴅ-glucopyranoside produces fluorescent colonies when exposed to long-wave UV radiation (365 nm) following cleavage of the fluorogenic 4-methylumbelliferyl moiety [167]. Chromogenic media uses the chromogen 5-bromo-4-chloro-3-indolyl-ɑ-ᴅ-glucopyranoside, which breaks down in the presence of oxygen to form the blue–green pigment 5,5′-dibromo-4,4′-dichloro-indigotin [168]. The sensitivity of chromogenic medium may be further enhanced through the use of two different chromogens. Restaino and Frampton [169] reported the use of R&F *Enterobacter sakazakii* chromogenic plating medium (ESPM) for *Cronobacter* detection from foods and environmental sources. This agar contains two chromogenic substrates (5-bromo-4-chloro-3-indoxyl-ɑ-ᴅ-glucopyranoside and 5-bromo-4-chloro-3-indoxyl-β-ᴅ-cellobioside), which form distinct blue–black colonies on agar and bile salts to inhibit the growth of gram-positive bacteria and antibiotics, thereby controlling *Proteus* and *Pseudomonas* growth.

### 8.2. PCR-Based Detection Methods

Conventional end-point and real-time polymerase chain reaction (PCR) methods were applied for detection of *Cronobacter* species from environmental sources, including powdered infant formula. Detection of *gluA*, *rpoB, ompA* and *dnaJ* are the most extensively evaluated methods; however, these methods have some limitations, including the limit of detection [172]. To overcome these limitations, PCR is often combined with additional methods designed to improve sensitivity and specificity, including capillary electrophoresis laser-induced fluorescence (CE-LIF) or immobilisation using zirconium hydroxide [177,178]. The CE-LIF can be used to detect the fluorescence at 520 nm generated using a combination of SYBR Green I and DNA fragments [178]. Zirconium hydroxide can be used to immobilise low numbers of *C. sakazakii* in PIF prior to DNA extraction and PCR analysis to improve sensitivity [177]. A third-generation PCR protocol was recently developed, which is called droplet-digital PCR. In this method, the sample is partitioned into thousands of smaller droplets, which allows the amplification of the target at the single-molecule level, providing more accurate and sensitive detection of low levels of pathogens. Using improved propidium monoazide as a dye, Lv and Gu [199] used single-cell droplet digital PCR to detect *Cronobacter* in infant food samples, reducing the detection time to 3 h with a limit in detection of 23 CFU/mL in pure culture.

### 8.3. Immunological-Based Detection Platforms

Immunological detection systems based on enzyme-linked immunosorbent assays (ELISA), which is a biochemical test that detects antibodies and antigens in samples, weredeveloped for *Cronobacter* species. These immunoassays were inexpensive and faster when compared to conventional culture-based techniques, with positive results being obtained in a period ranging from a few hours to two days. ELISA tests are generally considered to be broad spectrum in nature; however, more sensitive and robust immunoassays were developed for *Cronobacter* species, including sandwich assays, which use two antibodies to bind to two sites on the antigen. Sandwich assays were also shown to improve the limit of detection for *C. sakazakii* in artificially inoculated PIF compared to indirect ELISA assays [181,182]. Immunochromatographic test (ICT) strips combine thin-layer chromatography with conventional bio-affinity interactions using bioreactant-functionalised coloured particles as a detector and other bio-affinity partner immobilised as a capture probe in a test zone of the strip [184]. PCR-generated amplicons can be detected using an ICT strip, which eliminates the need for gel electrophoresis [200]. In this method, nucleic acid is first amplified via PCR, with the forward primer labelled with biotin and the reverse primer labelled with digoxigenin. Amplified DNA can then be applied to ICT, with the biotin labelled-end interacting with a carbon-neutravidin conjugate and the digoxigenin labelled end interacting with an anti-digoxigenin antibody. A positive *Cronobacter* species ICT strip results in the appearance of two grey–black lines in both test zones.

### 8.4. Biosensor-Based Detection Systems

Biosensors are analytical devices consisting of a transducer coupled with a biological element, which may be an enzyme, antibody or nucleic acid, that interact with an analyte to generate a measurable signal that is proportional to the amount of bioreceptor–analyte interaction. A number of biosensor-based assays were previously developed to detect *Cronobacter* species. Fluorescence in situ hybridisation (FISH) binds nucleic acid probes to specific target regions of DNA or RNA [187]. Gold nanoparticles can be directly conjugated with thiol-containing biomolecules, such as antibodies, allowing for the development of various immunosensing assays targeting *Cronobacter* species. These assays include lateral flow assays (LFA) [188], electrochemically coupled immunoassays [190], dynamic light scattering platforms [201,202] and polyethylene glycol (PEG)-coated gold nanoparticles conjugated with anti-*C. sakazakii* antibodies, which allow for detection using plasmon extinction spectroscopy [193,194]. Silica-coated magnetic nanoparticles (Si-MNP) can extract *Cronobacter* genomic DNA, which can then be captured using labelled 16S rRNA probes and exposed to ICT strips [185]. The sensitivity of this method is further enhanced using a sandwich complex, in which personal glucose meters (PGM) are combined with antibody-coated (Si-MNP) and antibody- and glucose oxidase-coated silica nanoparticles, which are then used to hydrolyse glucose. The linear relationship between the decrease in glucose concentration and the logarithm growth of *C. sakazakii* creates a simple, point-of-care immunoassay for detection.

Magnetic beads labelled with specific oligonucleotide probes can detect *Cronobacter* species in PIF [181]. Nuclear magnetic resonance (NMR) biosensors consist of a polyclonal rabbit anti-*C. sakazakii* antibody coupled to superparamagnetic iron oxide nanoparticles that can detect *C. sakazakii* in dairy samples [195,200]. Aptamers are short, single-stranded DNA or RNA oligonucleotides that can be used as bioreceptors and detection tools for *C. sakazakii* [197,198]. The detection time for *Cronobacter* in PIF can be reduced to 3 h through combining aptamer amplification with rolling circle amplification [198].

Matrix-assisted laser desorption ionisation–time of flight mass spectrometry (MALDI TOF-ms) can be used to detect ribosomal proteins of *Cronobacter* species [88]. These proteins generate a unique profile or fingerprint consisting of spectral peaks, which can be used to identify bacteria from a database. Impedance technology can measure changes in electrical conductivity arising from bacterial metabolism during growth, which can be combined with a commercially available RNA hydridisation assay (impedance-RiboFlow™) to detect *Cronobacter* species in PIF [203].

### 8.5. Whole Genome Sequencing-Based Approaches

Recently, whole genome sequencing (WGS) approaches were developed and used to detect *Cronobacter* in the built food production environment. Not only is WGS inexpensive, but it provides high resolution data, including serotype, virulence factors and information on antimicrobial resistance-encoding genes. The FDA launched a pilot program in 2012 called GenomeTrackr, which aimed to build a database of genomic sequence information and accompanying metadata (geographic location, source and date) to characterise food, environmental and clinical isolates in an effort to identify potential outbreaks in near to real time [204]. Initially focussed on the collection of *Salmonella* genomes, the database was expanded over the years to include additional foodborne bacteria, including *Listeria*, *Campylobacter* and *Cronobacter*. Field laboratories in the United States and around the world collect food and environmental samples for pathogen detection, with positive samples submitted for whole genome sequencing. The raw fastq files are then quality checked using GenomeTrackr’s internal QA/QC pipeline before the data are uploaded to the NCBI’s pathogen detection portal. Users may create their own Bioproject and curate their data using this online platform [205]. GenomeTrackr allows users to identify clinical clusters matching food/environmental isolates and understand the root cause of contamination along the farm-to-fork chain

## 9. Conclusions

Emergence of *Cronobacter* species focused attention toward accurately identifying isolates and determining their sources. Through understanding the difference in virulence capacity encoded by isolates from different sources, effective preventive actions could be fine-tuned, thereby minimizing potential food safety issues associated with PIF consumption in vulnerable consumers. Multiple genes associated with internal regulatory systems and environmental stress resistance were investigated. However, from a risk assessment view, challenges remain in measuring the prevalence of *Cronobacter* in different ecological niches. A few studies on dose-response models for *Cronobacter*-contaminated PIF consumption were previously performed. Therefore, the exposure assessment assumption could be biased. More in depth investigations and reliable data from dose-response models could better estimate risk associated with *Cronobacter*. Despite the development of novel detection methods for *Cronobacter* species, providing rapid, accurate, sensitive and lower cost compared to conventional methods, a few obstacles prevent these novel methods from being applied in a real environment.

## Figures and Tables

**Table 1 microorganisms-11-01379-t001:** A table showing currently recognised seven species within genus *Cronobacter* along with their corresponding bio-groups.

Cronobacter Species	Bio-Groups
*Cronobacter sakazakii* sp. *nov.*	2–4, 7, 8, 11 and 13
*Cronobacter malonaticus* sp. *nov.*	5, 9 and 14
*Cronobacter turicensis* sp. *nov.*	16
*Cronobacter muytjensii* sp. *nov.*	15
*Cronobacter condimenti* sp. *nov.*	1
*Cronobacter universalis* sp. *nov.*	Separate genomospecies
*Cronobacter dublinensis* subsp. *Dublinensis* sp. *nov.*	12
*Cronobacter dublinensis* subsp. *lausannensis* sp. *nov.*	10
*Cronobacter dublinensis* subsp. *lactaridi* sp. *nov.*	6

**Table 2 microorganisms-11-01379-t002:** A table summarising recalls, withdrawals and safety alerts for *Cronobacter* and *Enterobacter* species in EU [32] and US [33] from 2018 to 2021.

	*Cronobacter* spp. (EU)	*Cronobacter* spp. (US)	*Enterobacter* spp. (EU)	*Enterobacter* spp. (US)
Feed materials	0	0	5	0
Pet food	0	0	8	0
Compound feeds	0	0	3	0
Cereals and bakery products	1	0	1	0
Nuts, nut products and seeds	0	0	1	0
Other food products/mixed	1	0	0	0
Milk products, non-infant	0	1	0	0

**Table 3 microorganisms-11-01379-t003:** A table summarizing current detection methods designed for *Cronobacter* species.

Detection Methods	Detection Time	Detection Limits	Comments	References
**Culture-based methods**				
Non-selective enrichment	7 days	Not specified	Additional tests required for confirmation	[160,161,162,163,164]
Selective enrichment	4–5 days	Not specified	Supplementation with NaCl and incubation at 45°C improves selectivity for some strains	[165]
Differential enrichment	48 h	1 CFU in a 300 g sample	Used in conjunction with medium that incorporates a test for metabolism of ɑ-glucopyranoside	[78]
Fluorogenic media	24 h	Not specified	MUɑGlc is less specific for *Cronobacter* spp. than XɑGlc	[166,167]
Chromogenic media	24 h	Not specified	Breakdown of XɑGlc forms blue–green colonies	[116,168]
Dual chromogenic media	24 h	Not specified	Contains two chromogenic substrates to enhance sensitivity	[169]
**PCR-based methods**				
Conventional	24 h	1000 CFU/mL	Detection limits increase following enrichment	[170,171,172]
Real time	24 h	10 to 100 CFU/mL	Assays target the MMS operon (*rpsU*, *dnaG*, *rpoD*) or *ompA* gene	[173,174,175,176]
Duplex	24–30 h	3 to 16 CFU/mL	Sensitivity increases when combined with immobilisation techniques or capillary electrophoresis-laser-induced fluorescence detection	[177,178]
Droplet digital	3 h	23 CFU/mL	May detect VBNC cells when combined with Propidium Monoazide	[179]
**Immunological-based methods**				
ELIZA, INC-ELIZA and sandwich ELIZA,	10–36 h	1 cell per 25 g PIF to 6.3 × 10^4^ CFU/mL	Uses polyclonal and/or monoclonal antibodies specific for their target cell	[180,181,182]
Fluorescence-based liposome immunoassay	13 h	6.3 × 10^4^ CFU/mL	Liposomes tagged with antibodies specific for target cell	[183]
Immunochromatographic strip test	1–16 h	10 cells per 10 g–10^6^ CFU/mL	PCR amplicon is labelled with digoxigenin on one side and biotin on the other side, which enables detection	[184,185]
Immuno-blotting analysis combined with cross priming amplification	60–70 min	88 CFU/ mL–3.2 CFU/100 g PIF	16S-23S rDNA internal transcribed space (ITS) is amplified and analysed via BioHelix Express strip (BESt)	[186]
**Biosensor-based methods**				
Fluorescence in situ hybridisation (FISH)	12 h	1 CFU per 10 g PIF	Uses a peptide nucleic acid (PNA) to improve hybridisation	[187]
Gold nanoparticle-enhanced lateral flow immunoassay	3 h	10^3^ CFU/mL	Gold nanoparticles conjugate to capture antibodies at the detection zone	[188]
Electrochemical immunosensing assays	15 min—not specified	2 × 10^1^ CFU/mL–9.1 × 10^1^ CFU/mL	Uses graphene oxide/gold composite nanoparticles conjugated with anti-*C. sakazakii* antibodies	[189,190]
Immunomagnetic-resistance sensor	4–8 h and 30 min	2 cells per 10 g PIF–10^3^ CFU/mL	Immunomagnetic particle-bound bacteria are separated from a mixed suspension using a magnetic force and concentrated into a purified culture	[191,192]
Surface plasmon resonance	2–24 h	10 CFU/mL–30 CFU in 25 g PIF	PEG-grafted gold nanoparticles conjugated with anti-*C. sakazakii* antibodies bind to bacteria and are detected with plasmon extinction spectroscopy	[193,194]
Personal glucose meter (PGM)	90 min	4.2 × 10^1^ CFU/mL	PGM combined with antibody modified silica-coated magnetic nanoparticles and antibody and glucose oxidase-coated silica nanoparticles	[195]
Light scattering immunoassay	Not specified	51 CFU/mL	The scattering light intensity of silver-coated gold nanoparticles is used as a signal output for detection	[196]
Aptamers technology	3 h–2 days	33.3 CFU/mL–2.4 × 10^3^ CFU/mL	Uses ssDNA aptamers that bind to *C. sakazakii* with high affinity	[197,198]

## Data Availability

Not applicable.

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
