# Peer review of "Cronobacter Species in the Built Food Production Environment: A Review on Persistence, Pathogenicity, Regulation and Detection Methods"

_microorganisms, 2023, doi:10.3390/microorganisms11061379_

Round 1

Reviewer 1 Report

Mousavi et al  Well written and timely article.  Please consider my suggestions for change to the organization

I'm suggesting a re-organization to this manuscript.  It is well written but could be vastly improved with re-organization.

Because the majority of the readers would likely be interested in reading this article from the recent news surrounding outbreaks, consider rearranging the article, starting with a new introduction, then section L 243 to L -568.  Then go into the good detail you have on Cronobacter.  This I believe would be more interesting method of presentation to the majority of the readers

Author Response

Dear Reviewer,

Thank you for your positive feedback on our article. We appreciate your valuable suggestions for improving the organization of the manuscript. We have carefully considered your comments and have made the necessary amendments accordingly. Below is our response explaining the changes we made to address your concerns:

Based on your recommendation to prioritize recent news surrounding outbreaks, we have reorganized the article to provide a more engaging reading experience for the majority of our readers. The revised structure of the manuscript now follows the sequence you suggested.

 We have included section L 243 to L -568, as you suggested. This section delves into the topic with the necessary depth, providing readers with a comprehensive understanding of the subject matter. By placing this section after the updated introduction, we aim to capture the readers' interest right from the start and maintain their engagement throughout the article.

After covering the initial section, we transition into a detailed exploration of Cronobacter, as you mentioned. This portion of the article now follows the updated introduction and section L 243 to L -568. By including a dedicated section on Cronobacter, we are able to provide an in-depth analysis of this topic, which we believe will enhance the value of our research for our readers.

Once again, we sincerely appreciate your valuable input, as it has greatly contributed to improving the overall organization of our manuscript. We hope that the revised structure, incorporating your suggestions, will result in a more coherent and engaging article that effectively communicates the significance of our research.

Thank you once again for your time and expertise in reviewing our work.

Best regards,

Zeinab Mousavi

Reviewer 2 Report

The review has been designed according to the journal format . All sections have been written in a correct language. However, the authors must add data about the environmental factors affecting the production of powdered infant formula. Also the importance and nutritional value of PIF  must be added to the review

Author Response

Thank you for taking the time to review our manuscript titled "Cronobacter species in the Built Food Production Environment: A Review on Persistence, Pathogenicity, Regulation and Detection methods". We appreciate your feedback and suggestions for improving the paper.

Regarding your first comment, we agree that it would be valuable to include information about the environmental factors that can affect the production of powdered infant formula (PIF). We have revised the manuscript to include a section on this topic, which provides an overview of the key processing steps that can impact the safety and quality of PIF. 

In response to your second comment, we have also added a section to the manuscript highlighting the importance and nutritional value of PIF. This section discusses the key nutrients and benefits of PIF for infants, as well as some of the potential risks associated with its use.

Reviewer 3 Report

Specific comments

Line 36: Please remove – after 45-°C. Same for Line 40

Line 107: please delete e after induce

Line 109: References not italic and also add a fullstop

Line 111: Add a fullstop after species

Line 112: Please add of after in the range

Line 113: -value not italic same in the rest of the paper

Line 127: please remove fullstop after Cronobacter sakazakii

Line 162: bacteria names must be in Italic. Check also Line 167-167,173,248 and throughout the manuscript

Line 393: please add space after 70

No comments

Author Response

Thank you for taking the time to review our manuscript titled "Cronobacter species in the Built Food Production Environment: A Review on Persistence, Pathogenicity, Regulation and Detection methods". We appreciate your feedback and suggestions for improving the paper. We have amended all the changes suggested by you as below:

Line 36: Changes amended

Line 107: e was deleted

Line 109: References was changes and also fullstop was added

Line 111: fullstop after species was added

Line 112: of added after in the range

Line 113: change amended

Line 127:  fullstop after Cronobacter sakazakii was removed

Line 162: bacteria names re-written as italic Italic. Check also Line 167-167,173,248 and throughout the manuscript

Line 393:  space added after 70

Round 2

Reviewer 1 Report

Thank you for rearranging the article focusing on the majority of the readers’ interests around the incidence in the USA!  Good on revising the References, reporting on the serious consequences of Cronobater infections, making the point that PIF is not a sterile product, what consumers can do to minimize risks, that even international regulations could not eliminate these risks, D-values increased with lower water activity.

Check on eliminating blank pages 14-16

Reviewer 2 Report

It is o.k